# COVID-19 Mortality Rate and Its Incidence in Latin America: Dependence on Demographic and Economic Variables

**DOI:** 10.3390/ijerph18136900

**Published:** 2021-06-27

**Authors:** Javier Cifuentes-Faura

**Affiliations:** Department of Financial Economics and Accounting, Faculty of Economics and Business, University of Murcia, 30100 Murcia, Spain; javier.cifuentes@um.es

**Keywords:** COVID-19, death rate, population density, vulnerable employment, life expectancy

## Abstract

The pandemic caused by COVID-19 has left millions infected and dead around the world, with Latin America being one of the most affected areas. In this work, we have sought to determine, by means of a multiple regression analysis and a study of correlations, the influence of population density, life expectancy, and proportion of the population in vulnerable employment, together with GDP per capita, on the mortality rate due to COVID-19 in Latin American countries. The results indicated that countries with higher population density had lower numbers of deaths. Population in vulnerable employment and GDP showed a positive influence, while life expectancy did not appear to significantly affect the number of COVID-19 deaths. In addition, the influence of these variables on the number of confirmed cases of COVID-19 was analyzed. It can be concluded that the lack of resources can be a major burden for the vulnerable population in combating COVID-19 and that population density can ensure better designed institutions and quality infrastructure to achieve social distancing and, together with effective measures, lower death rates.

## 1. Introduction

A new coronavirus, known as COVID-19, reported in late December 2019 in Wuhan Province, China. The rapid spread of this virus has led to a global pandemic and an unprecedented major health, social, and economic crisis [1]. During the pandemic, many sectors of activity have been paralyzed. Some, such as commerce or tourism, are severely feeling the consequences of this crisis [2,3,4]. With the aim of reducing the number of cases and the spread of COVID-19, different countries have taken containment measures such as home confinement [5], mobility restrictions [6], or the closure of non-essential services [7]. Education has become primarily online [8,9,10], and most companies have opted for telecommuting [11,12].

The economic crisis resulting from COVID-19 is expected to be the most severe since the Great Depression, which originated in the United States in the 1930s with the collapse of the New York Stock Exchange and led to a lasting global economic crisis. The significant drop in GDP [13,14,15,16] forecasts a slow recovery out of the recession [17,18,19]. Economic reconstruction [20] is needed in many sectors to lessen or alleviate the devastating effects and crisis being produced by COVID-19 [21,22,23].

During the pandemic, the unemployment rate has risen sharply in all countries of the world [24,25,26,27,28]. There has been a sharp fall in the demand for goods and services and a disruption of global value chains [29,30,31,32], and many governments have put in place business supports by providing mortgage loans or deferring tax obligations [33,34]. They have also offered subsidies to the most vulnerable households or those at risk of poverty, with the aim of reducing the impact generated by the loss of income or unemployment [35] and have increased the budget allocated to public health [36] to hire more health personnel, purchase virus protection materials, and be able to offer greater care to those infected.

In 2021, with the arrival of several vaccines from different pharmaceutical companies, herd immunity is expected to be achieved as the majority of the population is vaccinated [37,38], which offers a glimmer of hope to the population to return to normality and begin some recovery.

South America is one of the areas most affected by this health crisis. Proof of this is the high number of cases and deaths during the pandemic [33,39]. Some demographic aspects may have had a significant influence on the number of deaths worldwide [40,41] and in South American countries [42,43,44]. Among the variables considered, are population density [45,46] as well as life expectancy or the proportion of the population in vulnerable employment. In this paper, we studied whether these demographic variables, together with the main indicator of a country’s wealth, GDP per capita, have a significant influence on the rate of covid deaths in Latin American countries.

There are few studies that analyze the economic and demographic variables that affect COVID-19, especially in Latin America, so with this work, we aim to fill this gap in the literature and to understand the effect that these variables have in order to optimize future decisions.

## 2. Literature Review

The recent pandemic produced by COVID-19 has triggered a great interest in knowing all the possible factors (economic, demographic, etc.) that may influence the number of infections and mortality caused by COVID-19. It is a very topical subject on which there is still not enough work, especially as regards Latin America.

When talking about demographic variables, one of the most representative and used is population density [47,48,49]. Population density is a measure of the population distribution in a given country or region that is equivalent to the number of inhabitants divided by the area where they live. This expression indicates the number of people per unit area, and its unit of measurement is inhabitants per km^2^ (hab/km^2^). Population density plays an important role in the study of emerging infectious diseases [50,51,52]. Since COVID-19 spreads when people are in close proximity [53,54], population density could be one of the aspects affecting the rate of spread and is one of the most interesting variables to study.

Residents living in areas with high population density, such as large cities, are more likely to come into close contact with others and, consequently, any contagious disease can be expected to spread more rapidly in dense areas [55]. However, several researchers concluded that the spread of COVID-19 is not related to population density. Hamidi et al. [55] concluded that counties with higher density in the United States have lower COVID-19 mortality rates than areas with lower density, possibly due to superior health care systems. In the same vein, Fang and Whaba [56] found that more densely populated Chinese cities such as Shanghai, Beijing, Shenzhen, Tianjin, and Zhuhai had far fewer confirmed cases per 10,000 people than other cities with lower population density in China. According to these authors, population density allows for economic development to ensure well-designed institutions, quality infrastructure, and effective interventions to achieve social distancing, making these regions more efficient against infectious diseases. Furthermore, Sun et al. [57] expound that population density cannot affect the spread of COVID-19 under strict closure policies.

However, not all studies published to date suggest that population density does not affect the mortality rate. According to Zamora Matamoros et al. [54], population density may influence positively or negatively, depending on the socioeconomic development achieved by cities. Coşkun et al. [58] determined that population density is one of the main factors involved in the spread of the virus in Turkey. Kadi and Khelfaoui [59] concluded that population density has a positive effect on the spread of COVID-19 in Algeria. Kodera et al. [60] showed that correlations between morbidity and mortality rates and population density were statistically significant (*p*-value < 0.05) in Japan. Ramirez and Lee [61] reported that population density was significantly and positively associated with the percentage of COVID-19 deaths in Colorado. Bhadra et al. [62] performed a correlation and regression analysis of COVID-19 infection and death rates at the district level and found a moderate association between the spread of COVID-19 and population density in India.

Life expectancy at birth may also influence the mortality rate related to coronavirus. Oksanen et al. [63] determined that, for the initial pandemic period of February to April 2020, life expectancy was positively associated with daily COVID-19 mortality in Europe. A study in Latin America determined that, during the first 90 days of the pandemic, life expectancy did not show an increased association with increased mortality rate [42]. According to Larochelambert et al. [64], higher mortality rates related to COVID-19 occur mainly in countries with longer life expectancy, which also have higher levels of GDP [64].

On the other hand, some authors state that low-income countries are more immune to COVID-19 [65], while others claim that countries with low income and low wealth have higher mortality [66], and that higher lethality may be related to the availability of health resources to cope with the pandemic [67]. COVID-19 particularly affects the most vulnerable population [68]. Being in a situation of vulnerability may increase the number of deaths related to COVID-19 [69], which is a major social challenge [70].

## 3. Data and Methodology

This paper analyzed, through a multiple regression model, whether demographic variables such as population density (*Density*), life expectancy (*Life expectancy*), and the percentage of the population with vulnerable employment (*Vulnerable*) have significant effects on the total number of deaths per million inhabitants (*Deaths*). We also included the economic variable of GDP per capita (*GDPpc*), closely related to demographic variables, which indicates the level of wealth of a country. This study was complemented by analyzing the influence of these explanatory variables on the total number of coronavirus cases per million inhabitants (*Cases*), in addition to their influence on the number of deaths. All South American countries for which data were available were selected. All data were obtained from information provided by the World Bank. A series of univariate analyses were also carried out to examine the variables used in the study.

To test the relationship between these variables, the following multiple linear regression models have been proposed:$$Deaths_{i}=\beta_{0\;}+\beta_{1\;}Density_{i}+\beta_{2}\;Life\;expectancy_{i}+\beta_{3}\;Vulnerable_{i}+\beta_{4}\;GDPpc_{i}+\varepsilon_{i}$$
$$Cases_{i}=\alpha_{0}\;+\alpha_{1}\;Density_{i}+\alpha_{2}\;Life\;expectancy_{i}+\alpha_{3}\;Vulnerable_{i}+\alpha_{4}\;GDPpc_{i}+\xi_{i}$$

## 4. Results

The number of deaths per million inhabitants varies considerably among the countries analyzed, as shown in Figure 1. The highest figures are found in Mexico (1342.41 deaths per million inhabitants), Peru (1324.04 deaths/million inhabitants), and Panama (1320.69 deaths/million inhabitants), and the lowest in Haiti (21.84 deaths/million inhabitants). The data shown were collected from the start of the pandemic (December 2019) until the beginning of February 2021.

The total number of confirmed cases also varies from country to country (Figure 1). As in the case of deaths, the data shown were collected at the beginning of February 2021. The highest incidence of cases per million inhabitants was found in Panama, followed by Argentina, Brazil, and Colombia, while the lowest incidence was found in Nicaragua and Haiti.

The highest GDP per capita of the countries analyzed is that of Panama (€14,143), and the lowest is that of Haiti (€1137). There is also great variability in population density (Figure 2), ranging from 11 inhabitants per square kilometer in Bolivia, 16 inhabitants per square kilometer in Argentina, and 17 inhabitants per square kilometer in Paraguay to 216 inhabitants per square kilometer in the Dominican Republic and 406 inhabitants per square kilometer in Haiti.

The percentage of the population in vulnerable employment is also quite high in South American countries (Figure 2), with Haiti in the lead (73%), followed by Bolivia (64%). Costa Rica (21%) and Argentina (22%) are at the opposite extreme. Finally, with regard to life expectancy, it is the highest in Costa Rica (80.3 years) and Chile (80.2 years) and the lowest in Haiti (64 years), with the average for the countries in the sample at 75.4 years.

A summary of the main descriptive measures of this study can be seen in Table 1. The different analyses were carried out using the SPSS statistical program (IBM, Armonk, NY, USA), and the results are shown in the following section.

The correlations between the different variables are shown in Table 2, with the highest variance inflation factor equal to 4.3, which shows the absence of multicollinearity problems.

The total number of deaths per million inhabitants showed a significant negative correlation with population density and a positive correlation with GDP per capita. In the countries analyzed, the higher the population density, the lower the life expectancy (negative correlation). Life expectancy appeared to be negatively and highly correlated with both vulnerable employment and GDP per capita. Finally, vulnerable employment showed a negative correlation with GDP per capita. Coronavirus cases and the number of deaths resulted to be highly positively correlated. Cases were also positively correlated with life expectancy and GDP per capita.

Multiple regression analysis (Table 3) revealed that all variables except life expectancy had a significant influence on the number of deaths in the South American countries analyzed. Population density and GDP per capita influenced at the 1% level, and the percentage of the population in vulnerable employment at the 5% level. The value of the standardized coefficients showed that GDP per capita is the variable with the greatest influence, followed by the percentage of vulnerable employment and population density.

Population density demonstrated a negative influence. The higher the population density, the lower the number of deaths per million inhabitants. These results are consistent with those of Hamidi et al. [55], who found that counties with higher density in the United States have lower COVID-19 mortality rates, and with those of Fang and Whaba [56] for China.

GDP per capita appeared to have a positive influence, i.e., those countries with higher GDP per capita have had more fatalities, as also shown by De Larochelambert et al. [64]. This is probably due to the high incidence of coronaviruses in more developed countries. The population in vulnerable employment also has a significant positive influence on the number of deaths. Countries with a higher percentage of vulnerable populations have had higher death rates. Lack of resources may be a major handicap for this population group in combating COVID-19, in agreement with Porcheddu et al. [67], Aquino-Canchari et al. [68] or Nemecio [69]. On the other hand, no evidence has been obtained that the life expectancy of the countries has an impact on the number of deaths due to COVID-19. The value of the standardized coefficients allows us to know the size of the effect of each dependent variable on the independent variable. The variable with the largest effect resulted to be GDP per capita. The effect sizes of population density and percentage of population in vulnerable employment were similar, although of different signs. Related to the size of the effect is the coefficient of determination, which measures the proportion of the dependent variable’s variation that is explained by the set of predictor variables, its range of variation being between 0 and 1. A coefficient of determination equal to 0.7 was obtained, meaning that 70% of the dependent variable is explained by the explanatory variables according to the linear model considered.

There could be an effect of differences in reporting between countries, if some countries report more reliably than others. This is why the analysis was carried out with “reliable” countries as a sensitivity analysis. The analysis was redone by eliminating the countries of Venezuela and Haiti, as they are the least transparent and perceived as the most corrupt according to the Corruption Perception Index, which could affect the reliability of their information. The conclusions obtained were similar, as shown by the *p*-values (Density = 0.02; Life expectancy = 0.44; Vulnerable = 0.09; GDPpc = 0.00).

To complement this study, we also chose to study the influence of the explanatory variables on the total number of cases of coronavirus, in addition to the total number of deaths. Another multiple linear regression was used for this purpose (Table 4).

The results showed, for mortality, that GDP per capita positively influences the incidence of confirmed COVID-19 cases. Life expectancy does not significantly influence it, and in this case, neither does population density. Vulnerable population is not significant at 5% but is significant at 10%. The size of the effect of GDP per capita is larger than both that of the percentage of the population in vulnerable employment and that of the effect on the death rate. Moreover, the percentage of variance explained by this model resulted to be 72%. As for COVID-19 deaths, a sensitivity analysis was performed, and the countries of Venezuela and Haiti were eliminated. The conclusions obtained were similar, as shown by the *p*-values (Density = 0.20; Life expectancy = 0.95; Vulnerable = 0.14; GDPpc = 0.01).

## 5. Conclusions

The rapid spread of COVID-19 worldwide has resulted in millions of infections and deaths worldwide, with South American countries being among the most affected.

Some papers showed the demographic characteristics and their influence on COVID-19 mortality rate in specific areas. However, none focused entirely on South America; therefore, this work will help to fill this gap in the literature.

By means of a correlation study and a multiple regression analysis, we analyzed whether population density, life expectancy, and the proportion of the population that is employed and vulnerable, together with the main indicator of a country’s wealth, the GDP per capita, have influenced the total number of deaths in Latin American countries.

Among the results, it stands out that countries with a higher number of inhabitants per square kilometers have had lower death rates, so that in these countries, population density is not a key factor that produces more deaths. Population density can enable economic development with strong institutions, high-quality infrastructure, and effective interventions that protect citizens from infectious diseases. It will also be necessary for government institutions to continue to put in place restrictive pandemic containment measures that will help reduce the rate of infection, and therefore deaths.

Countries with a high population density need not be more vulnerable to epidemics. Therefore, it can be concluded, along the lines of the work presented by Zamora Matamoros et al. [54], that population density may have a positive or negative influence, depending on the socioeconomic development achieved by cities. Once a certain threshold of population density is reached, better services can be provided to inhabitants, making it easier for them to stay at home and avoid unnecessary contact with others. It is important for the fight against COVID-19 that cities have adequate structures and means to combat the pandemic.

Countries with higher GDP per capita have had more deaths. The population in vulnerable employment also has a significant positive influence on the number of deaths. Countries with a higher percentage of vulnerable population have had higher death rates. Lack of resources may be a major handicap for this vulnerable population in combating COVID-19. On the other hand, there is no evidence that the life expectancy of countries can influence the number of deaths from COVID-19. To complement this study, in a second stage, the influence of the explanatory variables on the total number of cases of coronavirus per million inhabitants was analyzed by means of a multiple linear regression. The results showed that GDP per capita has had a positive influence on the incidence of confirmed cases of COVID-19, in line with the number of deaths. Efforts should be made to provide a support scheme for those who are at risk of social exclusion and are most vulnerable, as they have the least means to protect themselves from the coronavirus. Life expectancy showed no influence, and the difference, in this case, was that population density showed no influence either. On the other hand, as has been shown, it demonstrated a negative influence on the number of deaths.

Among the limitations of this work is that the data on deaths and COVID-19 infections are continually being updated, and this study analyzed them up to February 2021. In addition, it should be borne in mind that also the way the pandemic has been handled in different countries may have affected the results. Although the aim of this paper was to study the influence of demographic and economic variables, other factors that may influence COVID-19 mortality, such as the number of doctors per thousand inhabitants, the number of hospital beds, the proportion of people over 65 years of age, or the public health measures adopted in each country in response to the pandemic, can also be taken into account [42,71,72].

This study can be completed by taking into account other demographic and health factors that could help determine the death rate, such as chronic medical conditions that a patient may have.

## Figures and Tables

**Figure 1 ijerph-18-06900-f001:**
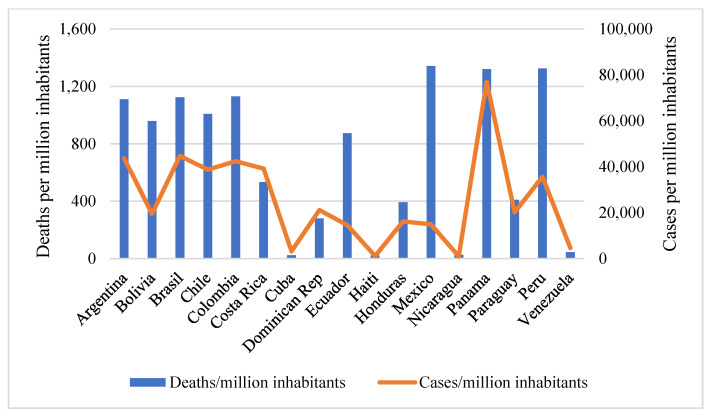
Number of COVID-19 deaths and cumulative confirmed cases of infection counted per million inhabitants. Source: own elaboration based on data provided by the World Bank. Data updated through February 2021.

**Figure 2 ijerph-18-06900-f002:**
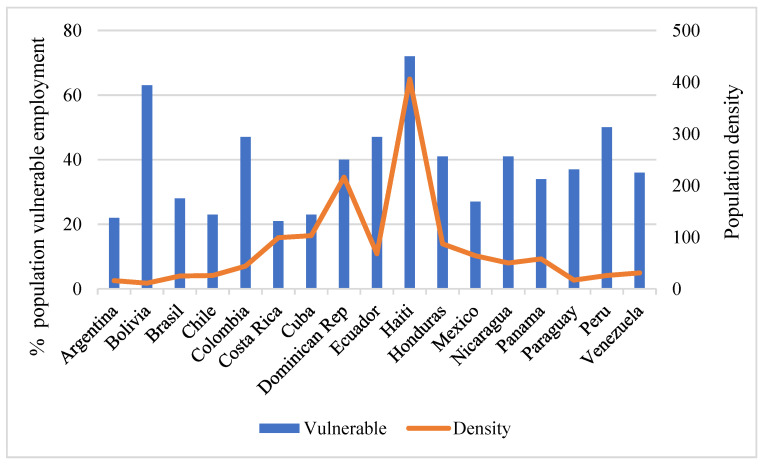
Percentage of population in vulnerable employment and population density. Source: own elaboration based on data provided by the World Bank.

**Table 1 ijerph-18-06900-t001:** Main Descriptive Measures.

Variables	Minimum	Maximum	Mean	Standard Deviation
Deaths	21.84	1342.21	701.46	506.09
Cases	969.67	76,911.35	25,731.21	20,351.21
Density	11	406	79.23	97.74
Health expectancy	64	80.3	75.43	3.86
Vulnerable	21	72	38.35	14.45
GDPpc	1137	14,143	6631.94	3842.83

**Table 2 ijerph-18-06900-t002:** Correlation analysis of the different variables.

	Deaths	Cases	Density	Life Expectancy	Vulnerability	GDP_pc_
Deaths	1					
Cases	0.733 **	1				
Density	−0.490 *	−0.354	1			
Life expectancy	0.421	0.544 *	−0.628 **	1		
Vulnerability	−0.152	−0.327	0.470	−0.759 **	1	
GDPpc	0.565 *	0.757 **	−0.267	−0.724 *	−0.674 *	1

* The correlation is significant at the 0.05 level. ** The correlation is significant at the 0.01 level.

**Table 3 ijerph-18-06900-t003:** Multiple regression model on the number of deaths from COVID-19 per million inhabitants.

Variables	Unstandardized Coefficients	Standardized Coefficients	*p*-Value
(Constant)	−1239.37 (3356.11)		0.718
Density	−3.475 ** (1.13)	−0.671	0.010
Life expectancy	−26.881 (43.29)	−0.205	0.546
Vulnerable	23.546 * (8.92)	0.673	0.022
GDPpc	0.130 ** (0.03)	0.987	0.002

Standard error in parentheses. ** Significant at 1% level. * Significant at 5% level.

**Table 4 ijerph-18-06900-t004:** Multiple regression model on confirmed cases of COVID-19 per million inhabitants.

Variables	Unstandardized Coefficients	Standardized Coefficients	*p*-Value
(Constant)	−20,672.122 (130,744.24)		0.877
Density	−69.302 (44.12)	−0.333	0.142
Life expectancy	−146.477 (1686.74)	−0.028	0.932
Vulnerable	702.621 (347.79)	0.499	0.066
GDPpc	5.427 ** (1.32)	1.025	0.001

Standard error in parentheses. ** Significant at 1% level.

## Data Availability

Not applicable.

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
