# Peer review of "COVID-19 Mortality Rate and Its Incidence in Latin America: Dependence on Demographic and Economic Variables"

_ijerph, 2021, doi:10.3390/ijerph18136900_

Round 1

Reviewer 1 Report

I have a couple of suggestions.

It could be added web link to a data source. You have specified only that the data obtained from the World Bank.

For me, interested question if whether of obesity has an impact on mortality? I guess that states didn’t do obesity measurements with covid patients and it is not easy to find data for that.

Reviewer 2 Report

In the manuscript "Covid-19 mortality rate and its incidence in Latin America: dependence on demographic and economic variables" by Cifuentes-Faura the author performed a univariable analysis and multivariable analysis of 4 demographic variables on country level (density, life expectancy, vulnerable, GCDpc), and Death per million and Cases per million in Latin American countries.

It is a good idea not to analyze countries worldwide, but to focus on Latin American countries that are more homogenous in many terms. The work is generally good.

I have several comments

General comments:
Be consistent with how COVID-19 is written. It appears in the manuscript also in lower cases Covid-19.

Is there an effect of reporting differences among countries? The author didn't relate to this important point. It may be OK to assume that, but it should be an explicit assumption. How would it change the results? Can the author estimates some countries have more reliable reports than others? If so, repeating the analysis with 'reliable' countries as a sensitivity analysis.

Another way for sensitivity analysis should be performed on data on a different period. For example, the first year of the pandemic or last three months (latter preferred), and make sure the results are consistent.

How the pandemic was handled in different countries, or in different continents and regions may also affect the outcomes in a complex manner. The author should comment on that.

GDPc positive correlation was also shown in Klinger et al. (https://www.mdpi.com/2076-393X/8/3/378/htm) who performed a multivariable analysis for 54 countries with ~20 variables.

The author should provide the complete table of data as a supplementary.

Specific comments:

Introduction
Not everyone is familiar with the "Great Depression". I advise writing a sentence or two about this economic period in the 30s.

The last sentence in the introduction "To date, it is not known that any work has been published on the demographic situation of these countries, and this is the great novelty of this study, apart from the interest 58 that this pandemic arouses among the research and scientific community." should be rephrased in a more modest way.

Literature Review
Don't use Ellipsis, but use 'etc.' in the parentheses.
First sentence: What do you mean by "... influence it". The subject is not clear, do you mean covid19 infections/ deaths/ spread, or something else.

In the last sentence of the second paragraph. "involved in the spread", you probably mean "affecting the rate of spread".

Change "Life expectancy at birth is a demographic variable that could also influence the coronavirus mortality rate." to "Life expectancy at birth may also influence mortality rate from coronavirus."

Data and methodology
First formula (Deaths), Life expectancy is missing index i.
Remove hat over Cases in the second formula.

"A study of correlations", probably should be replaced with "A series of univariable analyses"? And there there is no need to list the variables again, but state that the same list of variables was used for correlation analysis.

lines 134-139. The numbers are not a rate. As rate should be a number of death per million per unit of time. And it not clear enough whether these numbers are the commutative numbers ever since December 2019 up to February 1st, 2021.

Line 134 and on should not be part of the data and methodology section, but Results.

Figure 2. "...the data shown correspond to the beginning of February 2021", are the number of daily /monthly new cases? Or the Current number of confirmed cases? The cumulative number of confirmed cases?

Figure 2. Order countries in the same order as Figure 1, or in Descending order, but not alphabetically (unless the author reorders countries in Figure 1 alphabetical as well). Don't use a filled line plot, use bar char as Figure 1. Even better if the first two figures are merged into one figure with two panels.

Figure 3. This is not a suitable figure. As we don't compare countries based on their area, but density. Better to use a bar chart, where you can provide also the value.

Figure 4. Be consistent with the order of countries. And put all 4 first figures in one figure with different panels.

Table 2. Order variables so outcomes (death and cases) are next to each other in the top or bottom.
Why standard error has a single number in parentheses and not two, as in the first row?

Line 222: "The population in vulnerable employment also has a significant positive influence on the number of deaths." You mean negative, not positive, as the coefficient is -0.67

When comparing multiple regression of deaths and cases, doesn't show it is "quite similar". Only GDPpc found to be significant in both analyses and in the same direction. Vulnerable has the opposite effect (positive for cases, but not significant).

Reviewer 3 Report

First of all, I am grateful for the opportunity to review this paper. COVID-19 is an ongoing pandemic that has resulted in global health, economic and social crises. It is demonstrated that some demographic aspects, such as population density, as well as life expectancy or the proportion of vulnerable people, may have a significant influence on the number of deaths for COVID-19. In this context, the paper under review is aimed at investigating demographic variables, together with the main indicator of a country's wealth, GDP per capita, have a significant influence on the rate of covid deaths in Latin American countries.

The article is interesting and may provide important information for public health, but it must be improved.

Introduction: The authors should make it clear about what is the gap in the literature that is filled with this study? What is the contribution of the study to the literature? What are the implications of the study?

Methods: source of data and sampling procedure will benefit from more detail.

Statistical analysis: I suggest to insert a measure of the magnitude of the effect for the comparisons. Please consider to include effect sizes (for example the Cohens'd, if appropriate).

Discussion: I also suggest expanding. Emphasize the contribution of the study to the literature, the implications and recommendations based on previous experience. Limits section must be improved, the authors do not analyzed other factors influencing the burden and mortality of the diseases, such as adherence to preventive measures  (refer to to Roma P, et al How to Improve Compliance with Protective Health Measures during the COVID-19 Outbreak: Testing a Moderated Mediation Model and Machine Learning Algorithms. Int J Environ Res Public Health. 2020,17:7252).

English must be improved.

Round 2

Reviewer 2 Report

The manuscript was improved significantly.

A few comments:

Line 225: "pre-dictator", should be 'predictor'

Line 231: Which countries were considered "reliable"? How was it defined? What are the exact results? You can't write similar results, it must be quantified, so the reader should assess whether it's convincing.

Line 241: The results do not show that vulnerable employment influence the incidence of cases. The p-value is larger than 0.05, and this is the threshold usually used for significance.

Line 271,288 (and look for more): Change Covid-19 to upper case.

Be consistent with how you write: "GDPpc" or "GDP per capita".

Reviewer 3 Report

The paper was improved. Although - as reported by the authors - the aim of this paper is only to focus on the influence of demographic and economic variables, I strongly suggest to better discuss in the limits section the role of other factors influencing the burden and mortality of the diseases, in particular adherence to preventive measures referring to appropriate references, that were not considered in the analysis and may affect the results.
